# Sleep Is a Family Affair: A Systematic Review and Meta-Analysis of Longitudinal Studies on the Interplay between Adolescents’ Sleep and Family Factors

**DOI:** 10.3390/ijerph20054572

**Published:** 2023-03-04

**Authors:** Fabio Maratia, Valeria Bacaro, Elisabetta Crocetti

**Affiliations:** Department of Psychology “Renzo Canestrari”, Alma Mater Studiorum University of Bologna, Viale Berti Pichat 5, 40126 Bologna, Italy

**Keywords:** family, adolescents, sleep, systematic review, meta-analysis, longitudinal

## Abstract

Family is one of the primary socialization contexts influencing adolescents’ psychological health. In this regard, a crucial indicator of adolescents’ health is their sleep quality. Nevertheless, it is still unclear how multiple family factors (i.e., demographic and relational) are intertwined with adolescents’ sleep quality. For this reason, this systematic review with meta-analysis aims to comprehensively summarize and integrate previous longitudinal research investigating the reciprocal relation between demographics (e.g., family structure) and positive (e.g., family support) and negative (e.g., family chaos) relational family factors and adolescents’ sleep quality. Several search strategies were applied, and a final set of 23 longitudinal studies that matched the eligibility criteria were included in this review. The total number of participants was 38,010, with an average age at baseline of 14.7 years (SD = 1.6, range: 11–18 years). On the one hand, the meta-analytic results showed that demographic factors (e.g., low socio-economic status) were not related to adolescents’ sleep quality at a later time point. On the other hand, positive and negative family relational factors were positively and negatively related to adolescents’ sleep, respectively. Furthermore, the results suggested that this association could be bidirectional. Practical implications and suggestions for future research are discussed.

## 1. Introduction

Family is a dynamic context characterized by continuous changes due to the increasing diversity of its structure and challenges [1]. In adolescence, the family continues to represent one of the main socialization contexts [2,3]. Adolescents’ relationships with their family members play a crucial role in their development and adjustment [4]. Thus, multiple family factors and dynamics can promote adolescents’ psychological health and protect them in challenging developmental phases [5]. In this regard, a crucial indicator of adolescents’ health is their sleep quality.

Sleep is crucial for adolescents’ physical, cognitive, and psychological development [6]. Good sleep quality is conceptualized as a multidimensional construct, composed of satisfaction with sleep, alertness during waking hours, regular sleep schedule, a proper amount of sleep duration, and ease of falling asleep and returning to sleep [7]. Adolescents’ sleep is often a matter of concern since young people tend to report short sleep duration, irregular schedules, and poor sleep quality [8]. Since poor sleep quality has been linked to a plethora of negative outcomes (for a review, [9]), it is of utmost importance to understand which factors could affect youth’s sleep.

Adolescents’ sleep quality can be influenced by various family factors [10,11,12]. On the one hand, demographic characteristics (e.g., family socio-economic status) are essential antecedents of adolescents’ sleep. On the other hand, relational aspects of the family (e.g., quality of family relationships, family support) can either be antecedents or consequences of adolescents’ sleep quality [13,14]. In this respect, it is of paramount importance to gain a deeper understanding of how these demographic and relational family factors are intertwined with sleep quality in adolescence.

In this vein, the current systematic review with meta-analysis aims to comprehensively summarize the evidence collected in existing longitudinal studies about the relationship between family factors and adolescents’ sleep quality. Pointing out a distinction between demographic and relational aspects of the family context will enable us to address the core question: how do different family factors affect sleep quality in adolescence over time and vice versa?

### 1.1. Family Demographic Factors and Adolescents’ Sleep

Theoretical models focused on the etiology of healthy sleep, such as the socio-ecological system model [12], biopsychosocial and contextual model [11], and transactional ecological model [11] converge in highlighting that family factors play a primary role in influencing sleep quality in pediatric populations. One cluster of family factors theorized [15,16] to affect adolescents’ sleep quality is demographic factors. In this regard, the factors that have been considered as the most important ones are the family socio-economic status (e.g., household income, parents’ educational level) [17,18] and structure (e.g., single parents, have any siblings) [19].

Multiple indicators of family socio-economic status (SES), such as parents’ educational level, employment status, and financial well-being, have been related to adolescents’ sleep quality (for a review, [16]). Results of cross-sectional studies indicated that few economic resources, and parents with low educational level or that are both working are all factors related to adolescents’ irregular sleep schedules (e.g., later bedtime, earlier waketime) and shorter sleep duration (e.g., [20,21,22]). However, longitudinal studies suggested that these family SES indicators do not necessarily have detrimental effects on sleep quality over time (e.g., [19]).

When considering the impact of the demographic characteristics of the family on adolescents’ well-being, it is of utmost importance to also consider the increasing complexity of the structure (e.g., living with both or one parent, having any sibling), which characterizes modern families [23]. On the one hand, cross-sectional studies suggested that adolescents living with one parent were likelier than those living with both parents to report poor sleep quality and short sleep duration [24]. Furthermore, having one or more siblings could also result in adolescents’ poor sleep quality [25], and sharing the bedroom with other family members could lead to shorter sleep duration [26]. On the other hand, also for these family factors, longitudinal studies did not confirm that living with one parent or sleeping with any siblings negatively affects adolescents’ sleep in the long term [19]. Due to the discrepancy in these results, the role of family demographics in influencing adolescents’ sleep over time is still largely unclear.

### 1.2. Family Relational Factors and Adolescents’ Sleep

The second cluster of family factors that have been theorized to influence sleep quality [10,11,12] comprises family factors that tap into the quality of the relationships that adolescents have with their family members. In this vein, both positive (e.g., family support) and negative (e.g., family conflict) relational factors play a crucial role in adolescents’ sleep quality (for a review, [11]). Moreover, contrary to the demographic factors reviewed above, longitudinal studies not only indicated that family relational factors impact adolescents’ sleep over time but also suggested the possible presence of a reciprocal effect, according to which adolescents’ sleep can affect the quality of family relationships [27].

Positive family relational factors refer to a family system characterized by close, warm, and responsive relationships among its members [11,28]. In this sense, positive relational factors linked to adolescents’ sleep quality include positive relationships with parents, family support, monitoring and rules, and autonomy granting. Spending quality time with parents, being supported by them, and following bedtimes rules foster adolescents’ sleep quality in terms of long sleep duration and fewer sleep problems (e.g., [28,29,30]). At the same time, longitudinal evidence showed that adolescents’ sleep problems (e.g., insomnia symptoms) could reduce the quality of relationships with parents and increase problems at home (e.g., [14,27]). However, when considering other indicators of sleep quality (e.g., sleep duration, sleep efficiency), longitudinal results are still unclear, suggesting that specific aspects of adolescents’ sleep quality unequally influence family relational factors [31,32].

In some cases, negative relational aspects may outweigh the positive and protective factors of the family. In this sense, family-related stress events, high family demands, and problems in general impact adolescents’ well-being and sleep (for a meta-analysis, [33]). In particular, cross-sectional studies showed that high family stress and conflicts were associated with adolescents’ sleep problems, such as insomnia [33,34]. At the same time, adolescents perceiving high demands from family tended to sleep less at night [35]. Furthermore, a family environment characterized by great confusion and chaos was associated with adolescents’ poor sleep quality and shorter sleep duration [36]. However, longitudinal studies did not always find a lasting impact of negative family factors on adolescents’ sleep quality. They also pointed to the potential bidirectionality of this association, according to which adolescents’ sleep problems can exacerbate stress and conflict at home (e.g., [14,37,38]).

### 1.3. The Present Study

Empirical research highlighted how both demographic and relational aspects of family functioning influence sleep quantity and quality. Lower family SES and its complex structure could harm adolescents’ sleep. At the same time, family relational aspects could foster (e.g., parental support) or hamper (e.g., family conflict) adolescents’ sleep. However, if cross-sectional results offer a consistent picture, longitudinal studies suggest a more complex interplay between family factors and adolescents’ sleep quality. On the one hand, longitudinal evidence underscores that concurrent associations are not necessarily maintained over time. On the other hand, they also highlight that the interplay may be more dynamic and bidirectional.

Building upon the current state-of-the-art, this systematic review with meta-analysis aims to comprehensively summarize and integrate previous longitudinal research investigating the reciprocal relationship between the family context and adolescents’ sleep to address two primary goals: first, to understand how family demographics and relational factors can affect adolescents’ sleep quality differently over time; second, to examine how different facets of adolescents’ sleep quality can change family relationships over time.

## 2. Materials and Methods

This study was conducted following the PRISMA guidelines (Preferred Reporting Items for Systematic Reviews and Meta-Analyses; [39]). The PRISMA checklist is available in the Appendix A. This systematic review was preregistered in the PROSPERO database, registration ID: CRD42021281002. The current study is part of a larger project aiming to review longitudinal research studying the interplay between sleep quality and several proximal (e.g., peers; [40]) and distal (e.g., macro-context; [41]) factors in adolescence.

### 2.1. Eligibility Criteria

Following the PRISMA guidelines [39], specific eligibility criteria were defined. For the study characteristics, studies were eligible for the systematic review if (a) participants were adolescents from the general population aged between 10/11 to 18/19 years old; (b) the study design was longitudinal (with at least two assessments, such as two-wave longitudinal studies or daily diaries); (c) studies examined at least one aspect of family factors and one related to the sleep quality; (d) sleep could be measured with either objective (e.g., actigraphy, polysomnography) or subjective standardized measures (e.g., sleep diaries; questionnaires). Regarding the publication’s characteristics, journal articles and grey literature that can be retrieved through database searches (e.g., doctoral dissertations) were included to avoid selection biases and strengthen the methodological rigor of the systematic review [42]. Finally, no restrictions were applied based on the year or language of publication (when articles/dissertations were published in a language other than English, professional translators were contacted).

### 2.2. Literature Search

To systematically identify eligible relevant research published in peer-reviewed journal articles or available as grey literature, different search strategies were applied. First, several bibliographic databases were systematically searched until 23rd September 2021: Web of Science, Scopus, PsycINFO, PsycArticles, PubMed, MEDLINE, ERIC, ProQuest Dissertations and Theses, and GreyNet. In each database, the following combination of keywords was searched: (Sleep* OR insomnia* OR polysomnogra* OR REM OR actigraph* OR EEG* OR motor activity* OR circadian* OR chronotype*) AND (pediatr* OR paediatr* OR teen* OR school* OR adolescen* OR youth* OR young* OR child*) AND (longitudinal* OR prospective* OR follow up* OR daily* OR day-to-day OR wave*). Full query strings used in each database are reported in the Appendix A.

This main bibliographic search was complemented with additional search strategies. The websites of the journals deemed most likely to publish studies on the topic were searched, identifying them using the statistics of the previous search on Web of Science. The 15 journals in which most articles matching our search strategy had been published were identified (complete list of journals is reported in the Appendix A) to screen in-press articles (e.g., online first) that matched the eligibility criteria. Furthermore, conference proceedings from recent sleep-related journals were screened (*Journal of Sleep Research*, in which European Sleep Research Society Congress proceedings were published, and *Sleep Medicine*, in which the World Sleep Congress proceedings were published). The reference lists of the most relevant published systematic reviews and meta-analyses were checked (e.g., [43]; the complete list is reported in the Appendix A). Finally, the reference lists of included studies were screened to further identify relevant studies not initially found through the other search strategies (this search was performed at the end of the selection process). The searches and screening were run and managed on Citavi 6 software (V. 6.14, Swiss Academic Software, Wädenswil, Switzerland).

### 2.3. Selection of Studies

The results of the search strategies are reported in the PRISMA diagram (Figure 1). A total of 36,748 abstracts were identified, and 16,327 duplicates were removed. Two independent raters screened the remaining records (N = 20,421) independently and simultaneously. The percentage of agreement was substantial (Cohen’s Kappa = 0.81). Discrepancies were discussed with a third rater, and the final decisions were taken to reach an agreement among the three evaluators.

A total of 371 records were selected at this step. Next, the full texts were screened following the same procedure used for abstract screening (the agreement was high; Cohen’s Kappa = 0.61). In total, 23 studies were included in this systematic review.

### 2.4. Coding of Primary Studies

To extract relevant information from the selected primary studies, an Excel spreadsheet was prepared. All the included studies were coded independently and simultaneously by two independent raters (the percentage of agreement was 95%). Discrepancies were discussed with a third rater and solved among the three evaluators.

First, the characteristics of the publication were coded: type of publication (i.e., journal article or grey literature), year of publication, and the language of publication. Second, the characteristics of the studies were coded: funding sources (i.e., international funding, national funding, local funding, multiple funding sources); the number of waves of the longitudinal design; the interval between waves; the dimensions of each study; and the source of information used to evaluate them (i.e., self-reports, objective assessment). Third, the characteristics of the participants were coded: sample size, gender composition of the sample (% females), mean age, geographical location, and ethnic composition of the sample.

Finally, data necessary for effect size computations were extracted. Due to the high heterogeneity of the studies included, different effect sizes were coded (i.e., odds ratio, cross-lagged correlations, Spearman’s rho, beta coefficients) to address how family factors (demographic and relational) and sleep quality indicators were longitudinally related (see Section 2.5). If only standardized beta regression coefficients were reported, the correlation coefficients were estimated based on Peterson and Brown’s formula [44]. When data for effect size computations were not reported in primary studies, study authors were contacted by email to request missing data. In total, nine authors were contacted to obtain all (or part of) the necessary data for effect size computations. If authors did not answer the first request, three reminders (one every two weeks) were scheduled. Two authors replied by providing the requested data; five replied specifying that they could not provide the required data (e.g., they could not access the dataset anymore); and two did not respond to the request. For this reason, seven studies were excluded because there were insufficient data as indicated in the PRISMA diagram (full texts excluded because of missing data; Figure 1).

### 2.5. Strategy of Analysis

To address the research question, data related to family factors measured at one time point (e.g., demographic characteristics at T1) and sleep quality variables at a later time point (T2), or sleep quality variables at one time point (T1) and family factor variables at the following time point (e.g., family support at T2) were coded. When possible, the effect sizes were converted into Pearson’s correlations to compare the effects across studies and compute overall summary statistics through meta-analytic techniques. Pearson’s correlations were converted into Fisher’s Z-scores for computational purposes and converted back into correlations for presentation [45]. For ease of interpretation, correlations of |0.10|, |0.30|, and |0.50| are considered small, moderate, and large effect sizes, respectively [46]. Variance, standard error, 95% confidence interval, and statistical significance for each effect size were computed.

When at least three studies [47,48] were available on the same association, a meta-analysis was conducted using the software ProMeta3.0 to obtain an overall estimate. The random-effect model was used as a conservative approach to account for different sources of variation among studies (i.e., within-study variance and between-studies variance; [49]). Moreover, heterogeneity across studies was assessed with the Q statistic, to test if it was statistically significant, and the I2 to estimate it (with values of 25%, 50%, and 75%, respectively, denoting a low, moderate, and high proportion of dispersion in the observed effects that would remain should the sampling error be removed; [50]. Moderator analyses were used to test which factors can account for the heterogeneity [51]. Numerical moderators (such as the age of participants and time-lag between waves) were tested through meta-regression when at least three studies for each moderator level were available [48]. Finally, publication bias was examined through the visualization of the funnel plot (i.e., a scatter plot of the effect sizes estimated from individual studies against a measure of their precision, such as their standard errors). Without bias, the plot would be shaped as a symmetrical inverted funnel. However, since smaller or non-significant studies are less likely to be published, studies in the bottom left-hand corner of the plot are often omitted. The Egger’s regression method [52], which statistically tests the asymmetry of the funnel plot, was used, with non-significant results indicative of the absence of publication bias.

## 3. Results

### 3.1. Study Characteristics

Twenty-three studies were included in the systematic review. A summary of the characteristics of the included studies is reported in Table 1. In terms of year of publication, most of them (73.8%) were published between 2016 and 2021, and the remaining studies were published before 2016 (26.2%). The total number of participants was 38,010 (M = 1652.6, SD = 1678.8). Most samples were gender-balanced (the average percentage of females across samples was 52.5%; range 45.4–73.0%), and the average age of sample participants at baseline was 14.7 years (SD = 1.6, range: 11–18 years). Most studies reported one or multiple funding sources (83%). Regarding the context of the studies, most of the studies were conducted in the United States of America (65.2%) or Europe (8.7%); the remaining samples were from Australia [53], South Korea [19,54], Brazil [55], and Taiwan [56,57]. With regards to the study design, most of the studies (39.1%) included two time points, while the remaining studies included three or more time points (26.1%) and were daily studies (34.8%). The average time-lag between adjacent waves, excluding daily studies, was about one-and-a-half years (M= 16.6 months, SD = 18.8 months), ranging from 6 months to 7 years. Only one study used objective measures (i.e., actigraphy) to assess sleep variables; two studies used objective and subjective measures, while the remaining used only subjective measures.

### 3.2. The Longitudinal Influence of Demographic Factors on Adolescents’ Sleep Quality

Regarding the impact of different demographic family factors (i.e., educational level, household income, employment status, financial well-being, and family structure) on sleep quality indicators (i.e., sleep duration, sleep schedule, subjective sleep quality, and sleep disturbances) over time, 12 studies examined this link. In Table 2, all the effect sizes found in each study are reported. For a subset of six studies [18,31,58,59,60,61], it was possible to compute an overall effect size to estimate the relation between financial well-being and the educational level of parents and adolescents’ sleep quality at a later time point. The results (see Table 2) showed a non-significant effect (*r* = −0.02, *p* = 0.66). Heterogeneity was moderate and significant. However, the results were not moderated by the characteristics of the participants (i.e., mean age at T1, *B* = −0.25, *p* = 0.67) or of the studies (i.e., time-lag between waves, *B* = 0.18, *p* = 0.63). Furthermore, the visual investigation of the funnel plot suggested a low risk of publication bias that was statistically confirmed by a non-significant Egger’s test.

**Table 1 ijerph-20-04572-t001:** Characteristics of Studies Included in the Systematic Review.

Authors and Year	Sample Size Baseline	Sample Size Follow-Up	Mean Age (in Years)	% Female	Country	Number of Waves	Interval between Waves	Funding
Brodar et al., 2020 [62]	522	511	14.2	58.2%	U.S.	2	6 months	Yes
Chang et al., 2016 [55]	2491	n.a.	n.a.	49.4%	Taiwan	Daily	Day-to-day	Yes
Chang et al., 2019 [57]	2280	n.a.	n.a.	49.6%	Taiwan	10	12 months	Yes
Chiang et al., 2017 [37]	316	n.a.	16.4	56.9%	U.S.	Daily	Day-to-day	Yes
Fuligni et al., 2015 [17]	421	~337	15.0	50.0%	U.S.	2	12 months	Yes
Kim et al., 2020 [19]	2351	n.a.	n.a.	50.0%	South Korea	6	12 months	No
Meijer et al., 2016 [13]	650	493	13.3	50.4%	The Netherlands	3	12 months	No
Pasch et al., 2012 [18]	723	704	14.7	51.0%	U.S.	2	2 years	Yes
Peltz and Roger, 2019 [59]	193	n.a.	14.4	55.7%	U.S.	Daily	Day-to-day	Yes
Peltz et al., 2019 [31]	193	n.a.	15.7	54.5%	U.S.	Daily	Day-to-day	Yes
Peltz et al., 2020 [63]	193	n.a.	15.7	54.4%	U.S.	Daily	Day-to-day	Yes
Philbrook et al., 2020 [60]	252	214	15.7	53.0%	U.S.	3	12 months	Yes
Richardson et al., 2021 [53]	528	478	11.1	48.0%	Australia	3	12 months	Yes
Roberts et al., 2002 [27]	4175	3136	n.a.	48.8%	U.S.	2	12 months	Yes
Roberts et al., 2008 [14]	4175	3134	n.a.	48.8%	U.S.	2	12 months	Yes
Roberts et al., 2009 [32]	4175	3134	n.a.	48.8%	U.S.	2	12 months	No
Roberts et al., 2011 [38]	4175	3134	n.a.	48.8%	U.S.	2	12 months	No
Schäfer et al., 2016 [55]	5249	4563	n.a.	n.a.	Brazil	2	7 years	Yes
Sladek et al., 2019 [61]	209	n.a.	18.1	64.4%	U.S.	Daily	Day-to-day	Yes
ten Brink et al., 2021 [58]	381	381	14.4	49.0%	U.S.	Daily	Day-to-day	Yes
van den Einjdeen et al., 2021 [30]	2021	1422	13.8	45.4%	The Netherlands	2	12 months	Yes
Wang and Yip, 2020 [64]	256	256	14.7	73.0%	U.S.	Daily	Daily	Yes
Yoo, 2020 [54]	4335	n.a.	12.9	47.0%	South Korea	4	12 months	Yes

Note: With regards to the characteristics of the publication, all the studies were articles published in peer-reviewed journals and the English language.

Moreover, as for the additional studies that could not be included, two studies [17,19] considered only the employment status of parents, and the results showed that it was not associated with adolescents’ sleep duration but could affect their sleep schedule during the weekdays. When considering family structure, living with one parent or other relatives [19] and sharing the bedroom with one or more persons [55] was positively associated with adolescents’ longer sleep duration. At the same time, the presence of siblings was not associated with adolescents’ sleep [19]. Furthermore, when considering the financial situation, one study [57] found a significant association between economic stress of the family and adolescents’ poorer sleep quality over time, but this association was not found in the other two studies that considered the relation between family income and adolescents’ sleep duration [38,54].

### 3.3. The Interplay between Family Relational Aspect and Sleep Quality

Regarding the interplay between positive and negative relational aspects of family and different sleep quality indicators, 18 studies examined this relation. Only one study [62] evaluated this connection bidirectionally, with most of the included studies considering the effect of positive and negative relationships with family on adolescents’ sleep.

Of these, seven studies [13,17,30,53,54,62,63] evaluated the longitudinal impact of positive family relational factors on adolescents’ sleep quality. For a subset of five studies [13,30,53,62,63], it was possible to compute an overall effect size of this relation. Results, summarized in Table 3, showed a significant but small effect (*r* = 0.14, *p* < 0.001). Heterogeneity was small and significant. Results were not moderated by the characteristics of the participants (i.e., mean age at T1, *B* = 0.19, *p* = 0.83) or by characteristics of the studies (i.e., time-lag between waves, *B* = 0.18, *p* = 0.77). Moreover, the visual investigation of the funnel plot suggested a low risk of publication bias that was statistically confirmed by a non-significant Egger’s test.

Moreover, 10 studies evaluated the longitudinal impact of negative family relational factors on adolescents’ sleep quality over time, and for a subset of six studies [31,37,57,59,60,63] it was possible to compute an overall effect size of this relation. Results, summarized in Table 3, showed a significant but small effect (*r* = −0.08, *p* < 0.01). Heterogeneity was small, albeit statistically significant. It was not explained by the characteristics of the participants (i.e., mean age at T1, *B* = −0.24, *p* = 0.46). Moreover, the visual investigation of the funnel plot suggested a low risk of publication bias that was statistically confirmed by a non-significant Egger’s test.

Finally, six studies [14,27,32,55,62,64] evaluated the specific effect of sleep quality on family relationship aspects. For a subset of three studies [55,62,65], a meta-analysis could be conducted to obtain overall estimates of the longitudinal association between higher sleep quality at one time point (T1) and positive family relational variables at a later time (T2). Results, summarized in Table 3, showed a non-significant effect (*r* = 11, *p* = 0.10). Heterogeneity was small, albeit statistically significant. Moreover, the visual investigation of the funnel plot suggested a low risk of publication bias that was statistically confirmed by a non-significant Egger’s test.

To compute the overall meta-analytic summary related to the association between positive family relational factors at T1 and sleep quality at T2, the effect sizes of the studies were recoded so that higher positive family relations at T1 were related to higher sleep quality at T2.

To compute the overall meta-analytic summary related to the association between negative family relational factors at T1 and sleep quality at T2, the effect sizes of the studies were recoded so that higher negative family relations at T1 were related to higher sleep quality at T2.

To compute the overall meta-analytic summary related to the association between sleep quality at T1 and family relational aspects at T2, the effect sizes of studies were recoded so that higher sleep quality at T1 was related to higher positive family relations at T2.

Since Roberts et al. [14,27,32,38] only reported the computed odds ratio and confidence interval, it was not possible to include them in the meta-analytic calculations. Moreover, since Yoo [54] only reported the effect of change in family relational factors on sleep duration at T2, it was not possible to include it in the meta-analytic calculation.

## 4. Discussion

There is an increasing awareness of the importance of adequate sleep for adolescents’ daily functioning, and physical and psychological health. At the same time, the contexts in which adolescents are embedded shape their sleep schedule and influence their sleep quality and problems [65]. In this vein, the relationships with family members are one of the most crucial social relationships for adolescents, and they have important implications for their sleep quality [66]. To understand how family factors are intertwined with adolescents’ sleep quality (i.e., sleep duration and schedule, subjective sleep quality, and presence of sleep problems), it is crucial to consider, on the one hand, demographic factors (e.g., family socio-economic status) and, on the other hand, positive (e.g., family support) and negative (e.g., family chaos) relational factors. For these reasons, the current systematic review with meta-analysis aimed to extend prior knowledge on this topic, focusing on longitudinal studies that examined the interplay between family factors and adolescents’ sleep quality. Overall, most studies investigated the impact of family factors on sleep quality, highlighting that (a) family demographic factors were not associated with adolescents’ sleep quality indicators over time, and (b) negative and positive relational factors were positively and negatively associated with adolescents’ sleep quality indicators, respectively. In contrast, few studies evaluated how adolescents’ sleep quality was related to positive family relationships over time, showing a small but not significant longitudinal association.

### 4.1. The Impact of Family Demographic Factors on Adolescents’ Sleep

This systematic review examined the longitudinal association between family demographic factors and adolescents’ sleep, considering (a) family socio-economic status (SES) and (b) family structure. The overall effect pointed to a non-significant association (the effect size was close to zero). This evidence indicates an important difference between cross-sectional and longitudinal research. While this review suggests the lack of a longitudinal association between family demographic factors and adolescents’ sleep, prior cross-sectional studies underscored that adolescents’ sleep quality was related to the educational level of their parents, the income of their family, or its composition [20,21,22,24] although the association was generally small (for reviews, [16,67]). Considering all these aspects, it is worth reasoning about the importance of the temporal aspect of this association. For instance, something adolescents perceive as a stressful event or phenomenon (e.g., family economic stress events) in a given period may not be perceived similarly after a year. In this vein, family demographic factors could affect adolescents’ sleep in a certain period, but the same impact is not necessarily maintained over time since this association can undergo various changes.

At the same time, the literature about family demographics and adolescents’ sleep was rather heterogeneous since studies used different indicators (e.g., income, parents’ educational level), separately or combined, to assess the family socio-economic status. Thus, it was not possible to decompose the contribution of each aspect. Furthermore, studies that considered family structure were few. For these reasons, more studies are needed to understand the relation between family demographic factors and adolescents’ sleep.

### 4.2. The Effect of Positive and Negative Relational Factors on Adolescents’ Sleep

This systematic review considered the interplay between family relationships and adolescents’ sleep quality differentiating between positive and negative relational factors. Notably, both associations were found to be significant over time. The effect sizes were generally small but still meaningful, considering their longitudinal nature [68].

First, this review highlighted that positive family relational factors enhance adolescents’ sleep quality. In particular, warm parent-adolescent relationships, high family support, and parental monitoring were associated with adolescents’ better sleep quality [13,30,53,62,63]. This evidence is in line with theoretical models underscoring the centrality of family relationships for adolescents’ development [10,11,12]. Thus, the family context plays a crucial role in understanding adolescents’ psychosocial development [69].

Second, the results of the review underlined that family negative relational factors negatively impact adolescents’ sleep quality. In particular, a family context in which chaos, conflict, stress, and demands are highly present can decrease adolescents’ sleep quality, dysregulating their sleep schedule [31,37,55,59,60,63]. Together with the previous result, this gives us a broader picture, suggesting, on the one hand, that family can act as a protective factor when characterized by nurturing relationships; on the other hand, if negative relational aspects are not managed, then its protective capacity diminishes, increasing the chances of adverse consequences for adolescents’ health.

### 4.3. The Effect of Adolescents’ Sleep on Family Relational Factors

Although most studies focused on family factors’ impact on adolescents’ sleep, the present review also tackled effects in the other direction to understand whether adolescents with unhealthy sleep can influence their family context. In particular, sleep problems in adolescence are related to lower family support and more chances of having conflict at home [55,62,64]. However, the meta-analytic result of the association between adolescents’ sleep and family relationships was small and not significant, probably due to the limited number of studies that examined it. Moreover, only one study considered the impact of adolescents’ sleep on negative family relational factors, finding that insomnia symptoms were related to more family conflicts over time [55]. Overall, although the research on the impact of adolescents’ sleep on family relationships is still limited, it points to the possible presence of a bi-directional association. Thus, similarly to what was found for other adolescents’ problem behavior (i.e., internalizing and externalizing problems, [20]; for a review, [70]), it is possible to also observe an erosion of family relationships when children show sleep problems.

### 4.4. Limitations and Suggestions for Future Research

The results of this systematic review should be considered in light of some limitations. The first limitation concerns the heterogeneity of the reviewed literature, especially regarding the family factors taken into account. Since family is a complex system in which multiple demographical and relational factors are intertwined with adolescents’ psychosocial development, it is of utmost importance that future studies uncover the relative impact of each aspect to provide a comprehensive understanding of the dynamic influence exerted by family processes.

Second, although theoretically the association between family relationships and adolescents’ sleep may be bi-directional, most studies examined it only in one predominant direction, addressing the implication of family factors on adolescents’ sleep at a later time point. Only one study [62] considered the bidirectionality of the relationship and reported the influence that adolescents’ sleep quality has on family relationships and vice-versa. Therefore, future longitudinal studies are needed to disentangle how family relationships and children’s sleep quality influence each other, throughout adolescence, also differentiating among effects that may unfold in the short, medium or long term.

Third, most studies included in the review relied solely on self-report measures. This was the case for both family factors (i.e., only seven studies included parents’ reports; [17,31,37,53,59,60,63]) and sleep quality (i.e., only three studies objectively measured sleep parameters through actigraphy; [37,61,64]. Thus, social desirability and shared variance issues may have inflated the findings. At the same, considering different perspectives (e.g., accounting for both adolescents’ and parents’ views on family conflict; [71]) and methods (e.g., considering both the subjective perception of sleep duration and objective recording of it; [72]), it is crucial to uncover the complexity of the dynamic interplay between the family context and adolescents’ adjustment [70]. Thus, future research should integrate, on the one hand, objective and subjective assessments of adolescents’ sleep and, on the other hand, parents’ and adolescents’ points of view.

Finally, to better understand how (i.e., underlying mechanisms) family demographics and relational factors and adolescents’ sleep quality are related over time and for whom (i.e., moderations) this association is more robust, it is required to design longitudinal studies with multiple assessments (while only 26.1% of the studies in the current review included three or more time points). In this way, it would be possible to identify relevant mediations (e.g., family chaos T1 → perceived economic discrimination → adolescents’ sleep quality T3; [73]) and moderators (e.g., parents’ dysfunctional sleep-related beliefs; [59]) playing a role in the interplay between family factors and adolescents’ sleep quality. Thus, future studies examining these topics will better underline the mechanisms through which family factors and adolescents’ sleep are intertwined and clarify which are the protective factors for adolescents. This knowledge is of utmost importance to developing evidence-based interventions.

## 5. Conclusions

This systematic review with meta-analysis provided a comprehensive synthesis of longitudinal research on the relationships between family demographics and relational factors, and adolescents’ sleep quality. On the one hand, meta-analytic results showed a non-significant effect of demographic factors on adolescents’ sleep quality. On the other hand, the findings from the review also showed that positive (e.g., family support) and negative (e.g., family conflict) family relational factors are positively and negatively associated with adolescents’ sleep quality, respectively. Finally, the results also showed a small but non-significant association between adolescents’ sleep quality and family relational factors.

This review has important implications for the theoretical understanding of the interplay between the family systems and adolescents’ development while also highlighting a number of knowledge gaps in the existing literature that should be addressed in future research. Likewise, this review has important practical implications. Understanding how adolescents’ sleep problems and family factors are intertwined could underline the importance of interventions aimed at promoting protective family factors for adolescents’ health by working in two directions: on the one hand, educating about the importance that a positive and supportive family context has on adolescents’ sleep quality and thus on the overall health outcomes of all its members; on the other hand, improving sleep health can enhance a better quality of family relationships, thus promoting the development of a virtuous cycle.

## Figures and Tables

**Figure 1 ijerph-20-04572-f001:**
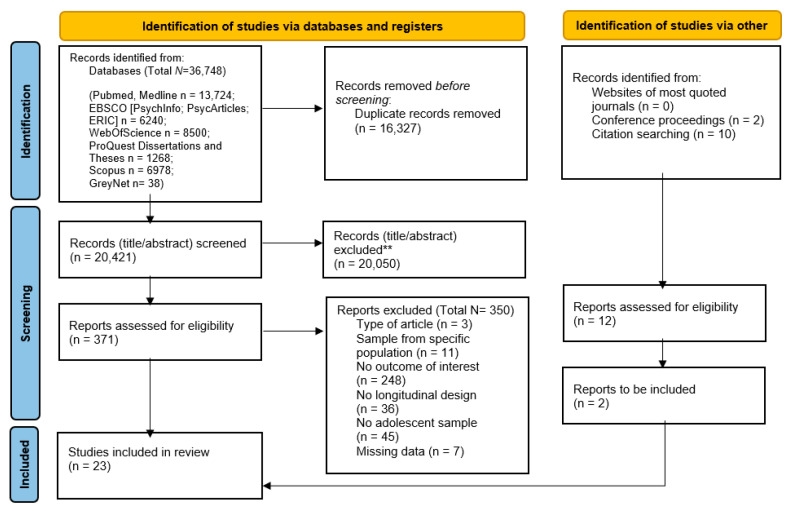
PRISMA 2020 flow diagram for new systematic reviews, which included searches of databases, registers, and other sources.

**Table 2 ijerph-20-04572-t002:** Longitudinal Associations Between Demographic Indicators of the Family and Adolescents’ Sleep.

Study	Family Demographic Category	Specific Family Demographic Indicator	Sleep Variables	Main Effect Reported	Effect Size Computed as Pearson’s Correlations	Main Findings
Chang et al., 2019 [57]	Family SES– Financial well-being	Family economic stress	Sleep quality (S)	*B* = −0.20 *** (0.03)		Family economic stress was negatively associated with adolescents’ sleep quality.
Fuligni et al., 2015 [17]	Family SES–Education	Parents’ educational level	Sleep duration (S)	*B* = −0.02 (0.01)		Adolescents’ sleep duration, waketime, and bedtime were not associated with their parents’ educational level.
			Waketime (S)	*B* = 0.01 (0.01)		
			Bedtime (S)	*B* = −0.00 (0.01)		
	Family SES–Work status	Parents working	Sleep duration (S)	*B* = 0.01 (0.01)		Adolescents’ sleep duration, waketime, and bedtime were not associated with their parents’ work status.
			Waketime (S)	*B* = −0.01 (0.01)		
			Bedtime (S)	*B* = 0.01 (0.01)		
Kim et al., 2020 [19]	Family SES–Education	Mothers’ educational level	Sleep duration (S)	(Weekdays) *B* = −13.97 *** (1.77) (Weekends) *B* = −17.33 *** (2.46)		Mothers’ high educational level (college degree) was negatively associated with adolescents’ sleep duration on weekdays and weekends. At the same time, mothers’ high educational level was positively associated with adolescents’ bedtime on weekdays and negatively associated with adolescents’ waketime on weekends.
			Bedtime (S)	(Weekdays) *B* = 11.88 *** (1.63) (Weekends) *B* = 3.74 (2.00)		
			Waketime (S)	(Weekdays) *B* = −1.33 (1.16) (Weekends) *B* = −13.04 *** (2.63)		
	Family SES–Education	Fathers’ educational level	Sleep duration (S)	(Weekdays) *B* = −12.58 *** (1.69) (Weekends) *B* = −13.30 *** (2.35)		Fathers’ high educational level (college degree) was negatively associated with adolescents’ sleep duration and positively associated with adolescents’ bedtime on both weekdays and weekends. At the same time, fathers’ high educational level was negatively associated with adolescents’ waketime only on weekends.
			Bedtime (S)	(Weekdays) *B* = 12.58 *** (1.55) (Weekends) *B* = 4.11 * (1.90)		
			Waketime (S)	(Weekdays) *B* = 1.02 (1.09) (Weekends) *B* = −9.018 *** (2.51)		
Kim et al., 2020 [19]	Family SES–Work status	Mothers working	Sleep duration (S)	(Weekdays) *B* = −1.74 (1.66) (Weekends) *B* = −1.55 (2.21)		Adolescents’ sleep duration, waketime, and bedtime were not associated with mothers’ work status (employed).
			Bedtime (S)	(Weekdays) *B* = −0.25 (1.48) (Weekends) *B* = 2.05 (1.78)		
			Waketime (S)	(Weekdays) *B* = −1.93 (1.12) (Weekends) *B* = −0.11 (2.31)		
	Family SES–Work status	Fathers working	Sleep duration (S)	(Weekdays) *B* = −19.88 *** (3.43) (Weekends) *B* = −12.83 ** (4.49)		Fathers’ working status (employed) was negatively associated with adolescents’ sleep duration and waketime during weekdays and weekends. At the same time, fathers’ working status was positively associated with adolescents’ bedtime only during weekdays.
			Bedtime (S)	(Weekdays) *B* = 8.23 ** (3.02) (Weekends) *B* = 2.36 (3.62)		
			Waketime (S)	(Weekdays) *B* = −10.07 *** (2.41) (Weekends) *B* = −10.51 * (4.63)		
Kim et al., 2020 [19]	Family SES–Work status	Both parents working	Sleep duration (S)	(Weekdays) *B* = −4.25 * (1.66) (Weekends) *B* = −1.88 (2.19)		Parents’ work status (both working) was negatively associated with adolescents’ sleep duration and waketime only on weekdays. There was no significant association between parents’ work status and adolescents’ bedtime.
			Bedtime (S)	(Weekdays) *B* = 0.52 (1.48) (Weekends) *B* = 0.38 (1.77)		
			Waketime (S)	(Weekdays) *B* = −3.54 ** (1.14) (Weekends) *B* = −2.02 (2.28)		
	Family SES– Financial well-being	Household income	Sleep duration (S)	(Weekdays) *B* = −0.00 *** (0.00) (Weekends) *B* = −0.00 *** (0.00)		Higher household income was negatively associated with adolescents’ sleep duration and positively associated with adolescents’ bedtime on both weekdays and weekends. At the same time, higher household income was negatively associated with adolescents’ wake time only on weekends.
			Bedtime (S)	(Weekdays) *B* = 0.00 *** (0.00) (Weekends) *B* = 0.00 *** (0.00)		
			Waketime (S)	(Weekdays) *B* = 0.00 (0.00) (Weekends) *B* = −0.00 * (0.00)		
Kim et al., 2020 [19]	Family Structure	Living with one parent	Sleep duration (S)	(Weekdays) *B* = 7.20 ** (2.74) (Weekends) *B* = 4.56 (3.71)		Living with one parent was positively associated with adolescents’ sleep duration on weekdays. At the same time, living with one parent was negatively associated with adolescents’ bedtime on weekdays but positively associated with it on weekends. Finally, living with one parent was positively associated with adolescents’ waketime on weekends.
			Bedtime (S)	(Weekdays) *B* = −4.24 * (2.47) (Weekends) *B* = 6.33 * (3.00)		
			Waketime (S)	(Weekdays) *B* = 1.80 (1.80) (Weekends) *B* = 9.73 * (3.90)		
	Family Structure	Living with others (e.g., grandparents)	Sleep duration (S)	(Weekdays) *B* = 9.28 *** (2.66) (Weekends) *B* = 2.27 (3.53)		Living with others (e.g., grandparents) was positively associated with adolescents’ sleep duration on weekdays. At the same time, living with others was negatively associated with adolescents’ bedtime on weekdays
			Bedtime (S)	(Weekdays) *B* = −6.13 ** (2.36) (Weekends) *B* = −0.61 (2.85)		
			Waketime (S)	(Weekdays) *B* = 2.53 (1.81) (Weekends) *B* = 1.20 (3.66)		
	Family Structure	Having any sibling	Sleep duration (S)	(Weekdays) *B* = −0.46 (2.38) (Weekends) *B* = 1.81 (2.21)		Having siblings was not associated with adolescents’ sleep duration, bedtime, or waketime.
			Bedtime (S)	(Weekdays) *B* = 2.16 (2.14) (Weekends) *B* = 2.20 (2.61)		
			Waketime (S)	(Weekdays) *B* = 1.68 (1.56) (Weekends) *B* = 4.33 (3.35)		
Pasch et al., 2012 [18]	Family SES–Education	Parents’ educational level	Sleep duration (S)	*β* = −0.02	*r* = −0.02 [−0.10, 0.06]	Parents’ high educational level was not associated with adolescents’ sleep duration.
Peltz and Roger, 2019 [59]	Family SES–Education and financial well-being	Parents’ educational level and household income	Sleep Quality (S)	*r* = 0.01	*r* = 0.01 [−0.13, 0.15]	Parents’ educational level and household income were negatively associated with adolescents’ sleep duration but not with adolescents’ sleep quality.
			Sleep duration (S)	*r* = −0.15 *	*r* = −0.15 * [−0.29, −0.01]	
Peltz et al., 2019 [31]	Family SES–Education and financial well-being	Parents’ educational level and household income	Sleep Quality (S)	*r* = 0.01	*r* = 0.01 [−0.13, 0.15]	Parents’ educational level and household income were negatively associated with adolescents’ sleep duration, but not with adolescents’ sleep quality.
			Sleep duration (S)	*r* = −0.17 *	*r* = −0.17 * [−0.31, −0.03]	
Philbrook et al., 2020 [60]	Family SES– Financial well-being	Income to needs	Insomnia symptoms (S)	*r* = −0.19 **	*r* = 0.19 ** [0.06, 0.32]	Household income, but not perceived economic well-being, was negatively associated with adolescents’ sleep quality problems.
		Perceived economic well-being		*r* = −0.14	*r* = 0.14 [0.01, 0.27]	
Roberts et al., 2011 [38]	Family SES– Financial well-being	Household income	Sleep deprivation (S)	(Middle vs. Lower income) OR [95% CI] = 1.39 [0.77, 2.49] (Higher vs. Lower income) OR [95% CI] = 1.25 [0.65, 2.40]		Family income was not related to adolescents’ sleep deprivation.
Schäfer et al., 2016 [55]	Family Structure	Number of persons sleeping per room at home	Sleep duration (S)	Males (Sleep with one person vs. Sleep with no one) *B* (95% CI) = 0.08 (−0.12, 0.28) (Sleep with two or more people vs. Sleep with no one) *B* (95% CI) = 0.33 ** (0.13, 0.53) Females (Sleep with one person vs. Sleep with no one) *B* (95% CI) = 0.22 ** (0.02, 0.42) (Sleep with two or more people vs. Sleep with no one) *B* (95% CI) = 0.67 *** (0.46, 0.87)		The number of persons who slept in the same room as the adolescent at 11 years was associated with longer sleep duration in girls at 18 years.
Sladek et al., 2019 [61]	Family SES–Education and financial well-being	Parents’ educational level and financial status	Sleep duration (O)	*r* = −0.02	*r* = −0.02 [−0.16, 0.12]	Parents’ education and financial status were not associated with adolescents’ sleep duration.
Ten Brink et al., 2021 [58]	Family SES–Education	Mothers’ educational level	Sleep quality (S)	*β* = −0.07 (−0.21, 0.06)	*r* = −0.07 [−0.17, 0.03]	Maternal educational level was not associated with adolescents’ sleep quality over time.
Yoo, 2020 [54]	Family SES– Financial well-being	Household income	Sleep duration (S)	(1997 Cohort) *β*= −0.01 (2000 Cohort) *β*= −0.03		Family income was not associated with sleep duration in adolescents from 1997 and 2000 cohorts.
Overall effect		k	ES [95% CI]	Q	I^2^	Eggers’ test
Family demographic characteristics T1 → Sleep Variables T2		6	−0.02 [−0.09, 0.05]	12.22 *	59.08	0.41

Note. S= Subjective; O = Objective; *r* = Pearson’s correlation (confidence intervals are reported between square brackets); OR: odds ratio with confidence intervals in square brackets; *β* = standardized regression coefficient and standard error estimate or confidence interval in parenthesis; 1997 = 1997-birth cohort; 2000 = 2000-birth cohort; k = number of studies; ES = effect size; Q = heterogeneity test; I^2^ = heterogeneity test. *** *p* < 0.001, ** *p* < 0.01, * *p* < 0.05. To compute the overall meta-analytic effect (based on [18,31,58,59,60,61]), the effect sizes of studies were recoded so that a higher level of family financial well-being and educational level at T1 was related to higher sleep quality parameters at T2. Sensitivity analysis conducted without the studies of Pasch et al. [18] and Ten Brink et al. [58], which were the only three that reported beta coefficients (converted into correlation through Peterson and Brown’s formula), indicated that the overall estimate was not significant. Since Yoo [54] only reported the effect of change in family demographical factors on sleep duration at T2, and Roberts et al. [38] only reported the odds ratio as the effect size, it was not possible to include them in the meta-analytic calculation.

**Table 3 ijerph-20-04572-t003:** The Interplay Between Positive and Negative Family Relational Aspects and Adolescents’ Sleep Quality.

Study	Family Relational Factors	Sleep Variable	Family Factors T1 → Sleep Variables T2 (Main Effect Reported)	Family Factors T1 → Sleep Variables T2 (Effect Size Expressed as Pearson’s Correlations)	Sleep T1 → Family Factors T2 (Main Effect Reported)	Sleep T1 → Family Factors T2 (Effect Size ExExpressed as Pearson’s Correlations)	Main Findings
Brodar et al., 2020 [62]	Family support	Insomnia symptoms (S)	*r* = −0.16 ***	*r* = 0.16 *** [0.07, 0.25]	*r* = −0.24 ***	*r* = 0.24 *** [0.16, 0.32]	Family support was bidirectionally negatively related to insomnia symptoms in adolescents over time.
Chang et al., 2016 [55]	Family interaction	Insomnia symptoms (S)			*r* = −0.09 ***	*r* = 0.09 *** [0.05, 0.13]	Adolescents’ sleep problems were negatively associated with family interaction. At the same time, adolescents’ sleep problems were positively associated with family conflict but not with parental support.
	Family support				*r* = −0.03	*r* = 0.03 [−0.01, 0.07]	
	Family conflict				*r* = 0.17 ***	*r* = −0.17 [−0.21, −0.13]	
Chang et al., 2019 [57]	Family dysfunction (activity, support, conflict)	Sleep quality (S)	*r* = −0.08 ***	*r* = −0.08 *** [−0.13, −0.03]			Family dysfunction was negatively associated with adolescents’ sleep quality.
Chiang et al., 2017 [37]	Daily parents’ demands	Sleep efficiency (S, O)	*r* = −0.05	*r* = −0.05 [−0.16, 0.06]			Daily parents’ demands and family-related stressful life events were not associated with adolescents’ sleep efficiency or duration over time.
		Sleep duration (S, O)	*r* = −0.11	*r* = −0.11 [−0.22, −0.00]			
	Family-related stressful life events	Sleep efficiency (S, O)	*r* = −0.03	*r* = −0.03 [−0.14, 0.08]			
		Sleep duration (S, O)	*r* = 0.06	*r* = 0.06 [−0.05, 0.17]			
Fuligni et al., 2015 [17]	Family support	Sleep duration (S)	*B* = 0.11 (0.03) ***				Family support was positively associated with adolescents’ sleep duration but not associated with adolescents’ waketime or bedtime At the same time, parents’ demands and family conflict were not associated with adolescents’ sleep duration, waketime, or bedtime.
		Wake time (S)	*B* = −0.05 (0.03)				
		Bedtime (S)	*B* = 0.04 (0.03)				
	Parents demands	Sleep duration (S)	*B* = 0.07 (0.07)				
		Waketime (S)	*B* = 0.00 (0.04)				
		Bedtime (S)	*B* = −0.01 (0.04)				
	Family conflict	Sleep duration (S)	*B* = 0.07 (0.04)				
		Waketime (S)	*B* = 0.03 (0.03)				
		Bedtime (S)	*B* = −0.01 (0.04)				
Meijer et al., 2016 [13]	Relations with parents	Bedtime (S)	*r* = −0.08	*r* = 0.08 [−0.01, 0.17]			Quality of parent-adolescent relations and high parental monitoring were positively associated with adolescents’ sleep quality, but not with bedtime. At the same time, autonomy granting was not associated with adolescents’ bedtime or sleep quality.
		Sleep quality (S)	*r* = 0.24 ***	*r* = 0.24 *** [0.16, 0.32]			
	Autonomy granting	Bedtime (S)	*r* = 0.03	*r* = −0.03 [−0.12, 0.06]			
		Sleep quality (S)	*r* = −0.08	*r* = −0.08 [−0.17, 0.01]			
	Parental monitoring	Bedtime (S)	*r* = −0.17 ***	*r* = 0.17 *** [0.08. 0.26]			
		Sleep quality (S)	*r* = 0.16 ***	*r* = 0.16 *** [0.07, 0.25]			
Peltz and Rogge, 2019 [59]	Pre-bedtime arguing	Sleep quality (S)	*r* = −0.16 *	*r* = −0.16 * [−0.30, −0.02]			Pre-bedtime arguing with parents was negatively associated with adolescents’ sleep quality but not with adolescents’ sleep duration.
		Sleep duration (S)	*r* = 0.06	*r* = 0.06 [−0.08, 0.20]			
Peltz et al., 2019 [31]	Parents’ use of inconsistent discipline	Sleep quality (S)	*r* = −0.03	*r* = −0.03 [−0.17, 0.11]			Family chaos was negatively related to adolescents’ sleep quality but not associated with adolescents’ sleep duration. At the same time, parents’ use of inconsistent discipline was not associated with adolescents’ sleep quality or duration.
		Sleep duration (S)	*r* = 0.01	*r* = 0.01 [−0.13, 0.15]			
	Family chaos	Sleep quality (S)	*r* = −0.19 *	*r* = −0.19 ** [−0.33, −0.05]			
		Sleep duration (S)	*r* = −0.06	*r* = −0.06 [−0.20, 0.08]			
Peltz et al., 2020 [63]	Parental rules (Bedtime)	Sleep duration (S)	*r* = 0.20 *	*r* = 0.20 * [0.06, 0.34]			Bedtime-related rules were positively associated with adolescents’ sleep duration. However, parent-child disagreement concerning bedtime and parental rules about screen time and caffeine use were not associated with adolescents’ sleep duration.
	Parental rules (Screen time)		*r* = 0.07	*r* = 0.07 [−0.07, 0.21]			
	Parental rules (Caffeine)		*r* = 0.01	*r* = 0.01 [−0.13, 0.15]			
	Parent-child bedtime disagreement		*r* = −0.06	*r* = −0.06 [−0.20, 0.08]			
Philbrook et al., 2020 [60]	Family chaos	Sleep quality problems (S)	*r* = 0.23 ***	*r* = −0.23 *** [0.36, −0.10]			Family chaos was positively associated with adolescents’ sleep quality problems.
Richardson et al., 2021 [53]	Parental monitoring (Technology)	Sleep duration (S)	*rs* = 0.16 *	*r* = 0.17 * [0.08, 0.25]			Parental control of technology was positively associated with adolescents’ sleep duration. At the same time, parental control of technology was negatively associated with adolescents’ insomnia in all three waves.
		Insomnia symptoms (S)	*rs* = −0.13 *	*r* = 0.14 * [0.05, 0.22]			
Roberts et al., 2002 [27]	Relations with parents	Insomnia symptoms (S)			(Moderate insomnia vs. Low insomnia) OR [95% CI] = 1.61 *** [1.31, 1.98] (High insomnia vs. Low insomnia) OR [95% CI] = 2.93 *** [2.31, 3.73]		Both moderate and high insomnia were associated with adolescents’ relational problems with their parents.
Roberts et al., 2008 [14]	Problems at home	Chronicity of one or more of insomnia symptoms (S)			OR [95% CI] = 2.42 * [1.86–3.16]		Adolescents’ chronicity of one or more insomnia symptoms was associated with more problems at home.
Roberts et al., 2009 [32]	Problems at home	Short sleep duration during weekdays and weekends (S)			OR [95% CI] = 1.29 [0.91, 1.84]		Short sleep duration during weekdays and weekends was not associated with problems at home.
Roberts et al., 2011 [38]	Mothers’ stress	Sleep duration (S)	(Middle vs. Lower stress level) OR [95% CI] = 1.13 [0.64, 1.98] (High-stress vs. Lower stress level) OR [95% CI] = 1.47 [0.79, 2.77]				Mothers’ stress was not associated with adolescents’ sleep deprivation.
Roberts et al., 2011 [38]	Fathers’ stress	Sleep deprivation (S)	(Middle vs. Lower stress level) OR [95% CI] = 0.81 [0.47, 1.38] (High-stress vs. Lower stress level) OR [95% CI] = 0.75 [0.41, 1.36]				Fathers’ stress was not associated with adolescents’ sleep deprivation.
Van Den Eijnden et al., 2021 [30]	Parental rules (Internet use)	Sleep quality (S)	*rs* = 0.12 ***	*r* = 0.13 *** [0.07, 0.18]			Parental rules about Internet and Smartphone usage during the hour before going to sleep were positively associated with adolescents’ sleep quality. At the same time, they were associated with adolescents’ earlier bedtime.
		Bedtime (S)	*rs* = −0.21 ***	*r* = 0.22 *** [0.17, 0.27]			
	Parental rules (Smartphone use)	Sleep quality (S)	*rs* = 0.10 ***	*r* = 0.10 *** [0.05, 0.16]			
		Bedtime (S)	*rs* = −0.26 ***	*r* = 0.27 *** [0.22, 0.32]			
Wang and Yip, 2020 [64]	Family support	Sleep quality (S, O)			*r* = 0.01	*r* = 0.01 [−0.11, 0.13]	Adolescents’ sleep quality was not associated with family support over time.
Yoo, 2020 [54]	Parental absence time after school	Sleep duration (S)	(1997 Choort) *β* = −0.04				Parental absence time after school, parental support, and parental monitoring were not associated with adolescents’ sleep duration in 1997 and 2000 cohorts.
			(2000 Choort) *β* = −0.10				
	Family support	Sleep duration (S)	(1997 Choort) *β* = 0.04				
			(2000 Choort) *β* = −0.02				
	Parental Monitoring	Sleep duration (S)	(1997 Cohort) *β* = 0.03				
			(2000 Cohort) *β* = 0.28				
Overall effect		k	ES [95% CI]	Q	I^2^	Eggers’ test	
Family Positive Relational Factors T1 → Sleep variables T2		5	0.14 *** [0.09; 0.18]	9.16 *	56.32	−0.41	
Family Negative Relational Factors T1 → Sleep variables T2		6	−0.08 ** [−0.12; −0.03]	7.10 *	29.54	−0.87	
Sleep variables T1 → Family Positive Relational Factors at T2		3	0.11 [−0.02, 0.23]	16.55 ***	87.91	1.95	

Note. *B* = Unstandardized regression coefficient and standard error estimate or confidence interval in parenthesis *β* = Standardized regression coefficient and standard error estimate or confidence interval in parenthesis; *r* = Pearson’s correlation coefficient; *rs* = Spearman’s Rho; OR = odds ratio and confidence interval in parenthesis; *k* = number of studies; ES = effect size; Q = heterogeneity test; I^2^ = heterogeneity estimate. *** *p* < 0.001,** *p* < 0.01, * *p* < 0.05.

## Data Availability

Data from previously published studies were retrieved and analyzed. Data sharing is not applicable to this article.

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
