# Peer review of "Sleep Is a Family Affair: A Systematic Review and Meta-Analysis of Longitudinal Studies on the Interplay between Adolescents’ Sleep and Family Factors"

_ijerph, 2023, doi:10.3390/ijerph20054572_

Round 1
Reviewer 1 Report
Thank you for the opportunity to review this manuscript. The manuscript describes a very well-conducted systematic review with results that should be of interest to a large audience, given the close relationship between sleep and health. Frankly, the authors present a high-quality systematic review: perfectly justified from previous research and the relevance of the topic, accurately conducted according to standard guidelines (i.e., PRISMA), and well-addressed by exploring the potential impact on the results of several factors. From my perspective, the manuscript seems ready for publication.
Author Response
Thank you for the opportunity to review this manuscript. The manuscript describes a very well-conducted systematic review with results that should be of interest to a large audience, given the close relationship between sleep and health. Frankly, the authors present a high-quality systematic review: perfectly justified from previous research and the relevance of the topic, accurately conducted according to standard guidelines (i.e., PRISMA), and well-addressed by exploring the potential impact on the results of several factors. From my perspective, the manuscript seems ready for publication.
RESPONSE. We thank the Reviewer for appreciating our work.
Reviewer 2 Report
great review of the studies looking at the interplay between sleep issues and family life factors in adolescents
this was very well written and comprehensive. My only suggestion is for one more edit as there were a few typos - e.g. lines 38 and 62
Author Response
Great review of the studies looking at the interplay between sleep issues and family life factors in adolescents this was very well written and comprehensive. My only suggestion is for one more edit as there were a few typos - e.g. lines 38 and 62
RESPONSE. We thank the Reviewer for appreciating our work and for highlighting these two typos. We have corrected them and proofread the entire manuscript.
Reviewer 3 Report
This is a very well written and thorough review and meta-analysis. The methodology is appropriate and the citations are thorough. The authors have correctly identified the limitations and areas for further research. I have no recommendations for improvement.
Author Response
This is a very well written and thorough review and meta-analysis. The methodology is appropriate and the citations are thorough. The authors have correctly identified the limitations and areas for further research. I have no recommendations for improvement.
RESPONSE. We thank the Reviewer for the positive evaluation.
Reviewer 4 Report
The manuscript provides a detailed review and meta-analysis of research examining the longitudinal association between demographic and family factors on adolescents’ sleep timing, duration, and quality. Overall, the review is thorough and the authors do a commendable job of walking the reader through each step of their process and findings. Please see minor comments below.
In Figure 1, there are 2 asterisks in the box “records (title abstract) excluded**”, but there is no description of what the asterisks signify in the figure caption.
At the beginning of the results section (lines 256-258), there is text that appears to be leftover from the template and needs to be removed.
On line 297-298, the authors state that 2 studies showed an effect of the number of family members on sleep duration, but no direction is provided. Are more family members associated with longer or shorter sleep duration?
In table 2, in the entry for Ten Brink, et al., in the main effect reported column, the brackets are inconsistent (open parenthesis and closed square bracket).
The sub-headings in section 4 are confusing. Is “Relation-Sleep” a play on the word “Relationship”? I think the beginning of the headings (“Where I come from”, “Relation-sleep”, “Ask for direction”) should be removed and the headings should just be as informative for the reader as possible.
Document S1 mentions checklist items that are located on pages 26-28 or page 29, but the manuscript is currently only 26 pages long. Please ensure that the table is consistent with the final manuscript document.
Author Response
The manuscript provides a detailed review and meta-analysis of research examining the longitudinal association between demographic and family factors on adolescents' sleep timing, duration, and quality. Overall, the review is thorough and the authors do a commendable job of walking the reader through each step of their process and findings. Please see minor comments below.
RESPONSE. We thank the Reviewer for positive evaluating our work and for the time and effort allocated to provide suggestions for further improving it. Below are our responses to each comment.
- In Figure 1, there are 2 asterisks in the box "records (title abstract) excluded**", but there is no description of what the asterisks signify in the figure caption
RESPONSE. Thank you for highlighting this. The two asterisks were leftover from the template of PRISMA checklist figure. We deleted them.
- At the beginning of the results section (lines 256-258), there is text that appears to be leftover from the template and needs to be removed.
RESPONSE. Thank you for highlighting this issue. We have removed the leftover.
- On line 297-298, the authors state that 2 studies showed an effect of the number of family members on sleep duration, but no direction is provided. Are more family members associated with longer or shorter sleep duration?
RESPONSE. We apologize for the unclarity. We have revised the statement, which now reads as follows (page 8 from line 296 to line 298):
« When considering family structure, living with one parent or other relatives [19] and sharing the bedroom with one or more persons [55] was positively associated with adolescents' longer sleep duration. At the same time, the presence of siblings was not associated with adolescents' sleep [19].»
- In table 2, in the entry for Ten Brink, et al., in the main effect reported column, the brackets are inconsistent (open parenthesis and closed square bracket).
RESPONSE. Thank you for highlighting this issue. We have removed the square bracket.
- The sub-headings in section 4 are confusing. Is "Relation-Sleep" a play on the word "Relationship"? I think the beginning of the headings ("Where I come from", "Relation-sleep", "Ask for direction") should be removed and the headings should just be as informative for the reader as possible.
RESPONSE. Thank you for highlighting this issue, and we apologize for the unclarity. We removed the beginning of each heading, and we do believe that the headings are now as informative as possible for the readers:
«4.1. The Impact of Family Demographic Factors on Adolescents' Sleep» (page 20, line 390)
«4.2. The Effect of Positive and Negative Relational Factors on Adolescents' Sleep» (page 21, line 413)
«4.3. The Effect of Adolescents' Sleep on Family Relational Factors» (page 21, line 433)
- Document S1 mentions checklist items that are located on pages 26-28 or page 29, but the manuscript is currently only 26 pages long. Please ensure that the table is consistent with the final manuscript document.
RESPONSE. Thank you for highlighting these issues. We have corrected the errors and updated the items' locations in the checklist.